# Meta-Reinforcement Learning reconciles surprise, value, and control in the anterior cingulate cortex

Tim Vriens[1], Eliana Vassena[2,3], Giovanni Pezzulo[1], Gianluca Baldassarre[1], Massimo Silvetti[1]*

**1** Institute of Cognitive Sciences and Technologies, CNR, Rome, Italy, **2** Behavioural Science Institute, Radboud University, Nijmegen, The Netherlands, **3** Donders Institute for Brain, Cognition and Behaviour, Radboud University, Nijmegen, The Netherlands

* massimo.silvetti@istc.cnr.it

## Abstract

The role of the dorsal anterior cingulate cortex (dACC) in cognition is a frequently studied yet highly debated topic in neuroscience. Most authors agree that the dACC is involved in either cognitive control (e.g., voluntary inhibition of automatic responses) or monitoring (e.g., comparing expectations with outcomes, detecting errors, tracking surprise). A consensus on which theoretical perspective best explains dACC contribution to behaviour is still lacking, as two distinct sets of studies report dACC activation in tasks requiring surprise tracking for performance monitoring and cognitive control without involving surprise monitoring, respectively. This creates a theoretical impasse, as no single current account can reconcile these findings. Here we propose a novel hypothesis on dACC function that integrates both the monitoring and the cognitive control perspectives in a unifying, meta-Reinforcement Learning framework, in which cognitive control is optimized by meta-learning based on tracking Bayesian surprise. We tested the quantitative predictions from our theory in three different functional neuroimaging experiments at the basis of the current theory crisis. We show that the meta-Reinforcement Learning perspective successfully captures all the neuroimaging results by predicting both cognitive control and monitoring functions, proposing a solution to the theory crisis about dACC function within an integrative framework. In sum, our results suggest that dACC function can be framed as a meta-learning optimisation of cognitive control, providing an integrative perspective on its roles in cognitive control, surprise tracking, and performance monitoring.

**Data availability statement:** All code for simulations used in the paper is available on GitHub at: https://github.com/AL458/RML-CogControl

**Funding:** T.V. has been funded by a PhD grant from the Consiglio Nazionale delle Ricerche (CNR, Italy) (https://www.cnr.it//), grant number DUS.AD016.135, and by the Ministero dell'Istruzione, dell'Università e della Ricerca (https://www.mur.gov.it/it), grant 'PRIN 2022, grant N° 20227MPSEH'. E.V. was supported by an Open Competition Xs grant (NWO 406. XS.04.129) from the Netherlands Organisation for Scientific Research (NWO). G.P. has been funded by the European Research Council under the Grant Agreement No. 820213 (ThinkAhead), the Italian National Recovery and Resilience Plan (NRRP), M4C2, funded by the European Union – NextGenerationEU (Project IR0000011, CUP B51E22000150006, "EBRAINS-Italy"; Project PE0000013, "FAIR"; Project PE0000006, "MNESYS"), and the Ministry of University and Research, PRIN PNRR P20224FESY and PRIN 20229Z7M8N. G.B. has been funded from 'European Union - NextGenerationEU - PNRR', MUR code IR0000011, CUP B51E22000150006, project 'EBRAINS-Italy - European Brain ReseArch INfrastructureS Italy. M.S. has been funded by the Ministero dell'Istruzione, dell'Università e della Ricerca (https://www.mur.gov.it/it), grant 'PRIN 2022, grant N° 20227MPSEH'. The funders had no role in study design, data collection and analysis, decision to publish, or preparation of the manuscript.

**Competing interests:** The authors have declared that no competing interests exist.

## Author summary

An important debate in cognitive neuroscience concerns the neural basis of cognitive control and the role of the anterior cingulate cortex (ACC). None of the current theoretical frameworks has succeeded as a unified theory accounting for all the neural and behavioural experimental data in this domain. In this study, we reanalysed previous neuroimaging data on ACC activity during cognitive tasks using a novel computational perspective: meta-Reinforcement Learning. We show that this computational framework can predict a variety of data on ACC function that, overall, could not be captured by any of the previous models. We propose that meta-Reinforcement Learning offers a unified theory of ACC cognitive and computational function, with a focus on its role in cognitive control.

## Introduction

Humans constantly face complex decisions, ranging from selecting one out of more available options (like choosing between an apple or a cupcake) to choosing the amount of cognitive and bodily resources we want to invest to achieve a goal (effort allocation). Formally, these decisional processes are aimed at solving a trade-off between minimizing the cost of investing cognitive (or physical) resources and maximizing the gain from investing a certain amount of resources [1–5]. In order to optimize this trade-off, it is sometimes necessary to expend some (cognitive) effort early to receive a larger reward later. In experimental psychology, the mental processes deputed to control the level of cognitive effort we deploy to achieve a goal are referred to as *cognitive control.* This process allows us to select more effortful options when we anticipate higher rewards from them. A typical example of cognitive control is the ability to inhibit habitual responses when these are not appropriate for the current situation (e.g., changing the route to your workplace when there are roadworks). The dorsal anterior cingulate cortex (dACC) and the surrounding cortical areas within the medial prefrontal cortex (MPFC) are known to be involved in cognitive control processes (e.g.,[6]), as well as in Reinforcement Learning (RL) and decision-making (e.g.,[7]). The computational mechanisms underlying these functions are still unclear, above all considering that their activation has been routinely found in many different experimental paradigms investigating several aspects of higher-order cognition [8–19]. Ironically, this area has been labelled as the Rorschach test for neuroscientists [20].

The most recent and effective attempts to find a unified theory of dACC function have led to two competing frameworks: the Expected Value of Control framework [21] (EVC) and the performance monitoring framework [8,22].

The EVC framework states that the dACC is involved in estimating the value of exerting *cognitive control* during a specific task and in selecting the optimal control signal, i.e., the control signal that maximizes the estimated value [21]. This framework provides a theoretical explanation for several studies that show dACC activity

during both anticipation of cognitively effortful tasks and the trade-off between the advantages of exerting cognitive control and its intrinsic cost [18,23–25].

The performance monitoring framework is based on RL and proposes that the dACC function is mainly aimed at performance monitoring by computing *prediction error* signals, resulting from the comparison between the expected and the actual outcome of actions. Prediction error signals are essential to update expectations about future action-outcomes associations. This theory was implemented in two independently developed models: the Predicted Response Outcome model (PRO) by Alexander & Brown [8] and the Reward Value and Prediction Model (RVPM) by Silvetti et al. [22] These models show that the combination of state-action-outcome expectations and the relative *surprise* signals (computed as the absolute value of prediction errors) was sufficient to explain many of the experimental findings about dACC function, from error detection to monitoring conflict between competing responses [26].

Both these frameworks bear significant merits in offering mechanistic and integrative explanations of dACC functions, but none of them can explain the full range of empirical findings. A growing body of literature reports activity in the dACC linked to cognitive control (or more generally during anticipation of cognitive effort), in keeping with the perspective of the EVC framework [21] (see also [27,28]). Given that in some of these cognitive control studies dACC activity is observed in the absence of surprise or prediction errors [3,24,29] or while controlling for all possible sources of surprise [30,31], these findings are not easily accommodated by the performance monitoring framework. Conversely, the performance monitoring framework provides a good account of the experimental findings documenting the role of the dACC in surprise coding during RL-based tasks [26], while the EVC framework does not account, by design, for these results. Furthermore, a recent fMRI study compared the predictions of the two frameworks during a speeded decision-making task, which involved both cognitive control and performance monitoring [32]. This study suggests that the dACC function could be parsimoniously explained by performance monitoring mechanisms (involving prediction and prediction error), without postulating additional cognitive control optimization mechanisms [33]. Finally, both of these theoretical frameworks have been similarly successful in providing a computational account of the role played by the dACC in decision-making processes during foraging [34], as alternatives to the foraging value theory (FVT [35]), which proposes that the dACC is involved in tracking the value of foraging to find new resources vs engaging in resources exploitation in the current context. The EVC perspective suggested that the original experimental findings supporting the FVT [35] were driven by the confound between the foraging value and the choice difficulty [17], while the performance monitoring perspective (instantiated by the PRO model [8]) suggested that the foraging-related dACC activity was instead due to the surprise from choosing either to forage or to engage in the current environmental context [36].

Taken together, the conflicting results of the above studies create a theoretical impasse, since no single account seems to be able to provide an integrative account of dACC function that fully explains its multifarious roles in both cognitive control and performance monitoring.

Here, we show that an alternative theoretical framework overcomes this impasse by accounting for the roles of dACC in both cognitive control and performance monitoring: The Reinforcement Meta-Learner model [37,38]. The RML belongs to a framework named meta-Reinforcement Learning (meta-RL), whose algorithms are able to adapt their internal parameters as a function of environmental challenges [39,40]. The RML combines some of the features of the EVC framework and the PRO model within the perspective of meta-RL, and in previous studies, it already provided a computational account of dACC function from both cognitive control and performance monitoring perspectives [37,38,41–43].

This study aims to directly compare the predictions of the RML model with those of the two other prominent frameworks (performance monitoring and EVC) in the specific experimental paradigms that most represent the current theory crisis. This direct comparison allows us to critically evaluate whether the meta-RL perspective can provide a unified theory about the role of the MPFC (and in particular of the dACC) in human cognition. We test the RML predictions in three different experimental paradigms whose empirical results, taken as a whole, are irreconcilable with both the EVC and the performance monitoring frameworks. The first paradigm is represented by the aforementioned speeded decision-making task,

which provided support for the performance monitoring framework [32]. The second paradigm is represented by a verbal working memory task (WM) [30,31,44], showing dACC involvement in cognitive effort independently from surprise, in keeping with the perspective of the EVC framework. Finally, the third paradigm is represented by a foraging task [35], where both the EVC and the performance monitoring frameworks were effective in predicting the dACC activity, proposing a more general interpretation of the dACC function compared to the domain-specific FVT [17,36].

We will show that the meta-learning framework proposed in the RML successfully predicts the MPFC (including the dACC) activity in all these three experimental paradigms. Based on these findings, we propose a solution to the theoretical impasse generated by the competition between the previous accounts of dACC function, explaining its central role in both monitoring and cognitive control within the unifying perspective of meta-learning.

## Methods

### Description of the RML model

The RML is an autonomous agent that learns and makes decisions (related to both motor behaviour and cognitive functions) to maximize net value, i.e., reward discounted by the cost of motor and cognitive effort (for a formal description, see [38] or S1 Text).

The model has been validated across multiple tasks and research domains [37,38,41,42,45], and was used in this work without parameter tuning (keeping all the parameters as in the original papers describing the model). From the neurophysiological perspective, the RML simulates a cortical-subcortical macrocircuit including the MPFC (including the dACC) and two brainstem neuromodulatory nuclei: the ventral tegmental area (VTA) and the locus coeruleus (LC) (Fig 1A). These regions are simulated using four computational modules: two for the MPFC, one for the VTA, and one for the LC [37]. Previous literature has shown a connection between these areas (e.g., [37,46,47]). Furthermore, the neurotransmitters produced by these brainstem nuclei, dopamine and noradrenaline, are known to affect decision-making (e.g., [37,48,49]) and modulate the dACC function [47].

The RML architecture assumes that the activity of the MPFC during decision-making can be explained by three different computations: the value of a decision (motor or cognitive), the surprise generated by the environmental feedback following this decision, and the level of cognitive control selected by the agent. Decision-making strategies relative to action selection and cognitive control levels (policies) are learned by agent-environment interaction, and based on approximate Bayesian learning implemented as Kalman filtering [50]. These three components of MPFC activity are processed by two separate modules (Fig 1B), an *action selection module (MPFC$_{act}$ module)* – dedicated to motor action selection – and a *boost selection module (MPFC$_{boost}$ module)* – dedicated to control optimization.

The MPFC$_{boost}$ module performs control over motor and cognitive functions, by upregulating or downregulating activity in the VTA and LC. When faced with a decision, the RML first decides on what amount of control to exert during the decision (termed '*boost*'). When the optimal boost level is selected, the MPFC$_{boost}$ module sends a signal to the VTA and LC modules, which in turn influence both the MPFC modules. As the optimal policy for selecting the boost signal is learned, and the boost signal itself influences the learning process, we define the search for an optimal boost policy as meta-learning. Boosting the LC module promotes effortful motor actions and enhances information processing in other brain structures (cognitive effort) while boosting the VTA module improves reward-related signals and learning from non-primary rewards. Boost, however, implies an intrinsic cost, and the boost module implements meta-learning by dynamically optimizing the trade-off between performance improvement and boosting cost. For the sake of simplicity, in the text, we will equate the boost signal to cognitive control, although the boost signal can also influence the motivational component of decision-making, via VTA modulation, aside from cognitive and physical effort via LC modulation.

The MPFC$_{act}$ module evaluates the options the agent has in the current environment based on expected effort and reward and selects the optimal action directed toward the environment. Action selection in the MPFC$_{act}$ module is influenced by the LC module, which promotes effortful actions, and by the VTA input, which influences the expected value of

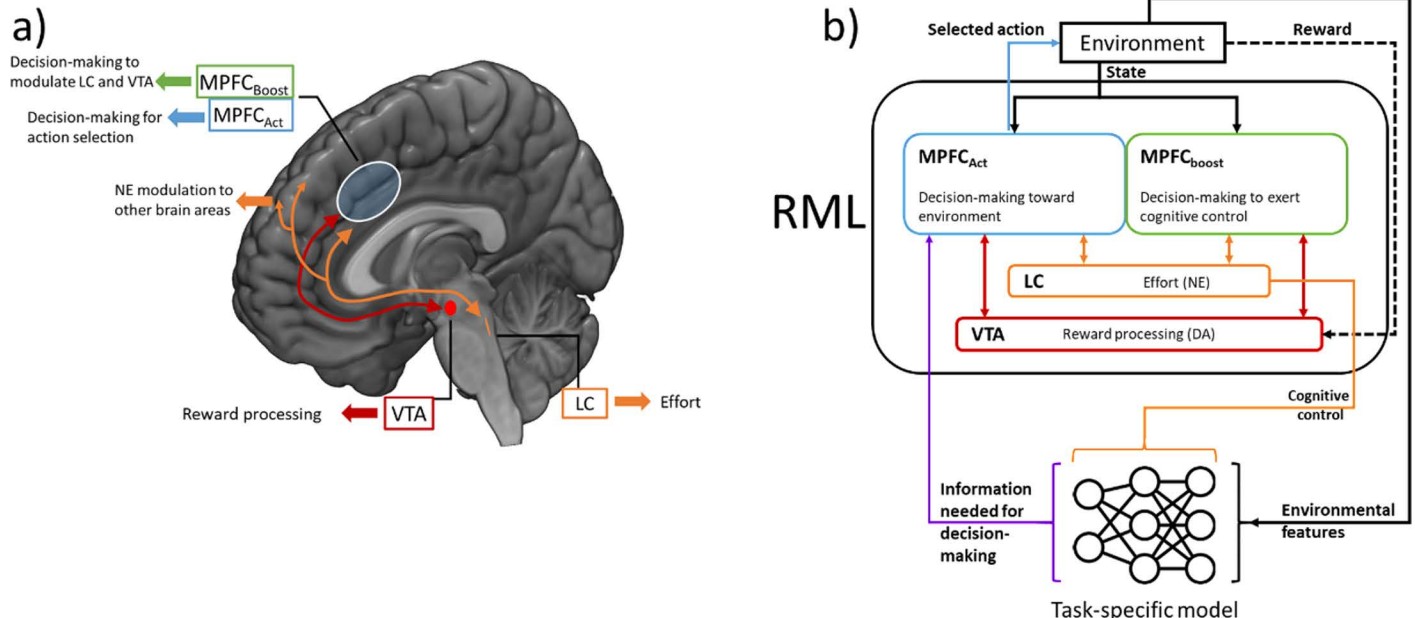

**Fig 1.** *Description of the RML model.* *a* Anatomo-functional mapping of RML modules. *b* Schematic representation of RML modules interactions. The agent consists of the RML and a task-specific model that provides the RML with specific functions necessary to execute a task. The RML optimizes the task-specific model via LC output (that can be interpreted as the cognitive control signal).

actions. In this way, the action selection and value learning of the MPFC$_{act}$ is indirectly modulated by the policy learned by the MPFC$_{boost}$ module (meta-learning for optimal action selection).

The RML works as a task-independent optimizer of motor and cognitive decision-making. This is possible because the RML can be connected to external, task-specific, modules (e.g., a deep neural network; Fig 1B). The activity of the external module is controlled via signals from the LC module, and its output is directed back toward the MPFC$_{act}$ module to provide the RML with the perceptual and/or motor features needed for a specific task.

**Interfacing the RML with task-specific external modules.** Based on the architecture shown in Fig 1B, we interfaced the RML with task-specific external modules to make it able to work as a generalized decision-making engine in different experimental domains. Below we describe an overview of the interfaces between the RML and the external modules in the three tasks we simulated in this work. For a detailed description, we refer the reader to the Supplementary Methods in the S1 Text.

**The RML model in the speeded decision-making task.** We simulated the speeded decision-making task as a set of time-constrained two-armed bandits, where the RML was asked to choose between several different stimuli -each linked to a specific reward value- presented in pairs in each trial so that there were trials where the value difference between the stimuli was small and trials where the value difference was large (detailed task description in the following paragraphs). In the speeded decision-making task, we connected the model to an external module simulating decisions by accumulating evidence over time, therefore enabling the model to simulate reaction times. To show the generalizability of our results, we replicated our simulations with two different well-known models as external modules: the drift diffusion model (DDM) as proposed by Ratcliff [51] and the dual attractor network (DAN) model by Usher & McClelland [52]. This ensured that the results could be attributed to the RML, and not to the specific external module used. In the main text, we describe the results of using the DDM as an external module. These were fully replicated with the DAN module, as described in detail in the S1 Text.

The integration of the DDM and the RML (Fig 2A) is inspired by Vassena et al. [32], where the authors fitted the drift rate and the distance of decision boundaries of a DDM to their behavioural data to demonstrate that the speeded

decision-making task relied on participants' cognitive control. In each trial, the DDM received two different signals from the RML as an input. The first was the difference in the expected value of the options ($\delta v$ in Fig 2A) from the MPFC$_{act}$ module. The value of each possible option was learned by the RML during a training session preceding the task. The absolute value of the $\delta v$ signal determined the DDM drift rate, while the sign of the $\delta v$ signal determined the drift direction toward one of the decision boundaries (representing the two available options, left vs right option) for that trial. In this way, a larger $\delta v$ absolute value resulted in higher accuracy and faster decision [32]. The second input from the RML to the DDM consisted of the LC signal, which modulated the distance of the decision boundaries. The higher the LC activity, the closer the boundaries, and the faster the decision. As the LC output is controlled by the boost signal from the MPFC$_{boost}$ (cognitive control), the latter can therefore modulate the decision speed as a function of trial type. Once the DDM reached one decision boundary, or no decision was made within the response time, this information was passed to the MPFC$_{act}$ module that implemented the option selection; finally, both the MPFC modules updated the expected value for the selected option based on the environmental feedback (reward).

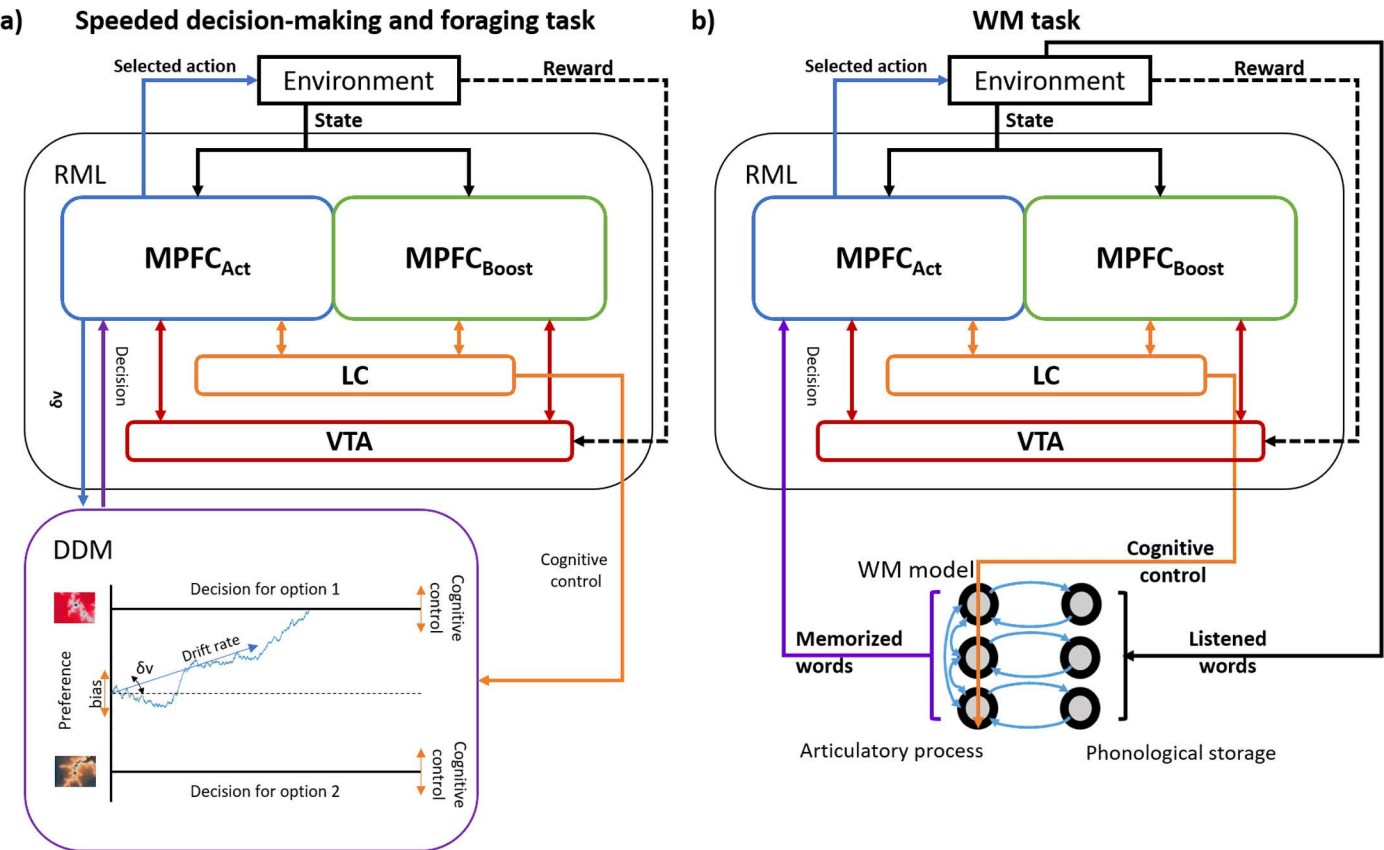

**Fig 2. *RML-external modules interactions in different tasks.* a:** *RML-DDM interaction during the speeded decision-making task and the foraging task. The RML received input from the environment (about rewards and environmental states), and controlled a task-specific module (the DDM), which helped in task execution. δv: difference in the expected value of the two options (different fractals for the speeded decision-making, "forage" or "engage" for the foraging task), whose absolute value determined the drift rate, while its sign determined the drift direction (up or down). The LC output modulated the decision boundaries, influencing the decision time. In the foraging task only, the LC output additionally influenced the bias of the DDM towards the "engage" option (human propension for "engaging" [17,35]). This bias was set to 0 for all trials in the speeded decision-making task. **b:** RML-cRNN interaction during the execution of the verbal WM task. The LC output modulated the gain of the neural units in the articulatory process layer, improving words retention in WM.*

**The RML model in the WM task.** The WM task consisted of a type of matching-to-sample task, where, in each trial, the model was asked to retain in memory a set of words and decide – after a delay- if a target word belonged to the previously presented set (detailed task description in the following paragraphs). To execute this task, the RML was connected to a task-specific model that simulated verbal WM functions (Fig 2B). This model consisted of a dual-layer competitive recurrent neural network (cRNN), inspired by the FROST network [53]. The input layer of the cRNN encoded the words presented in each trial (each unit encoded one word), working as a phonological storage. The output layer retained the words during the delay period, thanks to recurrent connectivity with the input layer. This layer simulated the articulatory process and also compared the memory content with the target word. Precision of words retention decreased with increasing WM load (number of words to be remembered) due to lateral inhibitory connections in the output layer. The RML improved words retention by gain modulation of the output layer, via LC output. The MPFC$_{Act}$ made the decision about words-target matching based on the linear combination of the cRNN output (see Supplementary Methods in S1 Text for details).

**The RML model in the foraging task.** In this task, the RML was asked to choose –in each trial- between exploiting the current reward patch ("engage" option) and foraging, i.e., exploring the opportunity provided by other patches ("forage" option; detailed task description in the following paragraphs). In the case of "engage" selection by the model, a specific two-armed bandit was presented (randomly extracted from the current patch), leading to a second stage of decision-making, where the bandit was played. A similar architecture as the one used for the RML-DDM in the speeded decision-making task was used for this experimental setting, with the exception that the foraging task included a bias term, which was modulated by the LC signal as well (Fig 2A, see also Supplementary Methods in S1 Text). The bias term was set to promote the "engage" option by default, simulating the human propension for this choice[17,35]. All the other features of the RML-DDM interaction were the same as those described for the speeded decision-making task. Given this architecture, an upregulation of the LC signal implied both a faster RT (decreasing the distance of the decision boundaries) and a facilitation to select the "forage" option, by discounting the bias term for the "engage" option.

## Task descriptions

**Speeded decision-making task.** We administered to the RML a version of the speeded decision-making task proposed in Vassena et al. [32]. In each trial, the agent was asked to select one among two options (representing fractal images in Fig 3A). In the original work by Vassena et al. [32], participants were asked to choose between two pairs of figures. Here we presented to the model two single options to reduce the implementation complexity of the simulations, without changing the original task structure. The left and right options were independently set to yield a reward equal to an integer between 2 and 7 (six reward levels, with two fractal images for each of the six reward levels). All possible combinations of the left and right reward were considered, for a total of 36 possible trial types (balanced by side). The goal of the agent was to select the best option (in terms of expected reward) in the shortest time possible. Once the RML selected the option, a reward corresponding to the fractal value was delivered to the model. In order to model the time pressure component of the task and incentivize the RML to respond as fast as possible, we implemented both a linear devaluation of the reward as a function of reaction time (S13 Equation, in S1 Text) and a response deadline (14000 DDM cycles). This solution captures the fact that human participants were instructed to respond as fast and as accurately as possible and that no rewards were given in case of excessively long RT [32]. The difficulty of each trial was a function of the value difference between the presented options: large value differences led to faster and more accurate choices than small value differences. Before the task execution, the model was trained to the value of each of the stimuli (see also Supplementary Methods in S1 Text). Simulation results are from the average of 200 simulations (participants). During each simulation, each of the possible trial types was presented 54 times, for a total of 1944 trials for each simulated participant.

**Working memory task.** In this simulation, we administered a WM task like the one used in Engström et al. [31] (Fig 4A), to the RML. This task exemplifies the role of the dACC in cognitive effort. During each trial, the RML was

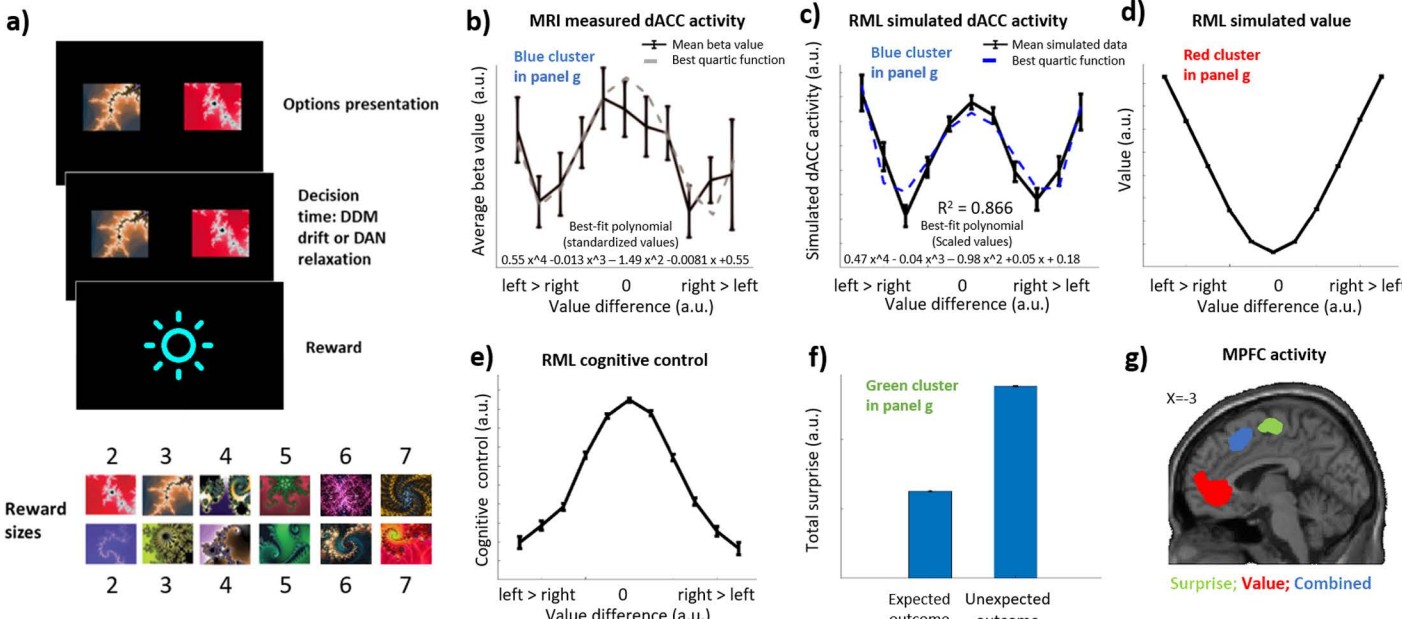

**Fig 3. Speeded decision-making task simulation. *a:*** *Task layout. The RML was presented with two possible choices (fractal images). Each image was assigned a hidden reward value ranging from 2 to 7 (bottom fractals list). The agent's task was to choose the image with the higher reward, and it had to do this as quickly as possible. We tested the RML on 36 different combinations of reward values for the two images. After the RML made its choice, it received the reward associated with the selected image.* ***b:*** *The MRI results from Vassena et al. [32] (adapted). The dACC activity is shown in the black line, while the grey, dashed line shows the best fitting quartic function to this data.* ***c:*** *The dACC activity as simulated by the RML (black line), and the best fitting quartic function (blue, dashed line). This activity is the sum of the value (panel d) and the cognitive control (panel e).* ***d:*** *The value component of the RML activity.* ***e:*** *Cognitive control signal (RML boost) as a function of stimuli value difference.* ***f:*** *RML surprise-related activity.* ***g:*** *Activation clusters within the MPFC (based on data extracted from the figures from Vassena et al. [32]). Blue: dACC activation as a mixture of value and cognitive control, RML prediction in panel c; Red: vMPFC value-based activation, RML prediction in panel d; Green: mid-cingulate activation relative to average surprise, RML prediction in panel* ***f.***

exposed to either 1, 4, 6, or 8 words, generating four different difficulty levels. After a delay of 10s, the model was presented with a target word that matched one of the memorized words in 50% of trials. The model's goal was to indicate whether the target word matched one of the words presented before (details in Supplementary Methods in S1 Text). In this task, results are from the average of 15 simulations (participants). For each simulated participant, 90 trials were presented to the model for each of the possible four difficulty levels (total of 360 trials).

**Foraging task.** The foraging task was based on the tasks proposed by Kolling et al. [35] and Shenhav et al. [17] (Fig 5A). The RML performed a sequence of trials, each randomly drawn from 16 different contexts (reward patches). Each of these environments contained 8 possible reward patches to exploit (consisting of 8 different two-armed bandits with different rewards associated with the optimal arm). In each trial, the RML was shown a compound cue indicating both the proposed bandit to play and the context where the bandit was randomly extracted from. The contexts differed from each other in the average value of the optimal arm of the bandits they proposed, so that some contexts provided on average more valuable bandit games than others. Selecting the "engage" option resulted in transiting to the following trial stage where the model executed the two-armed bandit indicated by the cue. Selecting the forage option resulted in showing a cue indicating the different options available in the context, followed by a transition to the initial state, where a random different bandit from the current context was proposed. The trial continued until the RML selected the engage option, or opted for the forage decision 20 times consecutively. Finally, a response deadline of 2000 DDM cycles was introduced, after which the trial was aborted. Here, differently from the speeded decision-making task, we implemented no devaluation of the reward as a function of reaction time. Before the foraging task, the RML was trained to learn the values of all the

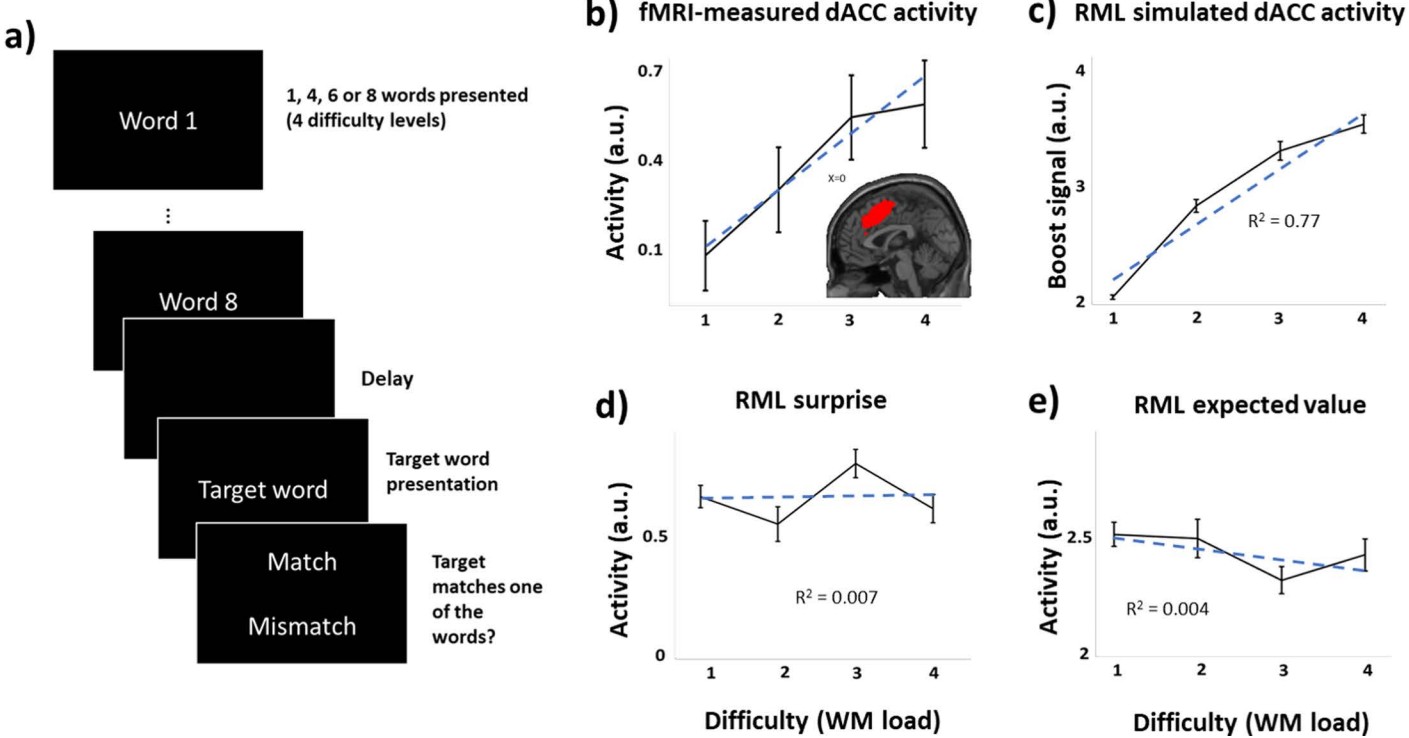

**Fig 4. Verbal WM task simulation. *a.*** *Setup of the verbal WM task. During each trial, the RML was exposed to either 1, 4, 6 or 8 words, generating four different difficulty levels. After a delay of 10s, the model was presented with a target word that matched one of the memorized words in 50% of trials. The model's goal was to indicate whether the target word matched one of the words presented before.* ***b.*** *fMRI results from Engstrom et al. [31] showing dACC activity (red cluster in subpanel) as a function of WM load (difficulty levels 1-4), dashed line shows linear fitting.* ***C.*** *Boost signal from RML as a function of WM load, dashed line shows linear fitting.* ***d-e.*** *RML surprise and expected value (average of MPFC$_{Act}$ and MPFC$_{Boost}$) as a function of WM load. Dashed lines show linear fitting.*

different bandits in all the different contexts (see Supplementary Methods in S1 Text for more details). Results are from the average of 40 simulations (participants). Each simulation consisted of 18 blocks, where each block contained all initial compound cues twice (a total of 4608 trials per simulated participant).

## Results

### Speeded decision-making task

In this simulation, we aimed to show that the RML can simulate the dACC activity during the speeded decision-making task used by Vassena et al. [32] (Fig 3A). In their study, Vassena et al. found that the human dACC activity followed a W-shaped function of the difference of value between the two options (a quartic function with a positive leading coefficient, see Fig 3B).

As expected, our simulation shows that the RML predicts the W-shaped activity pattern exhibited by the human dACC (Fig 3C and blue cluster in Fig 3G). Using the Akaike information criterion (AIC) [54], we found that the simulated dACC activity follows a quartic pattern with a positive leading coefficient rather than a quadratic pattern (Aikake weight for a quartic function > 0.999), the same reported in Vassena et al. [32]

The RML simulates the W-shaped dACC activity as the sum of two different neural signals evoked by the cue onset: the expected net value (expected reward discounted by expected costs, Fig 3D) and the cognitive control signal (boost signal in the RML, Fig 3E).

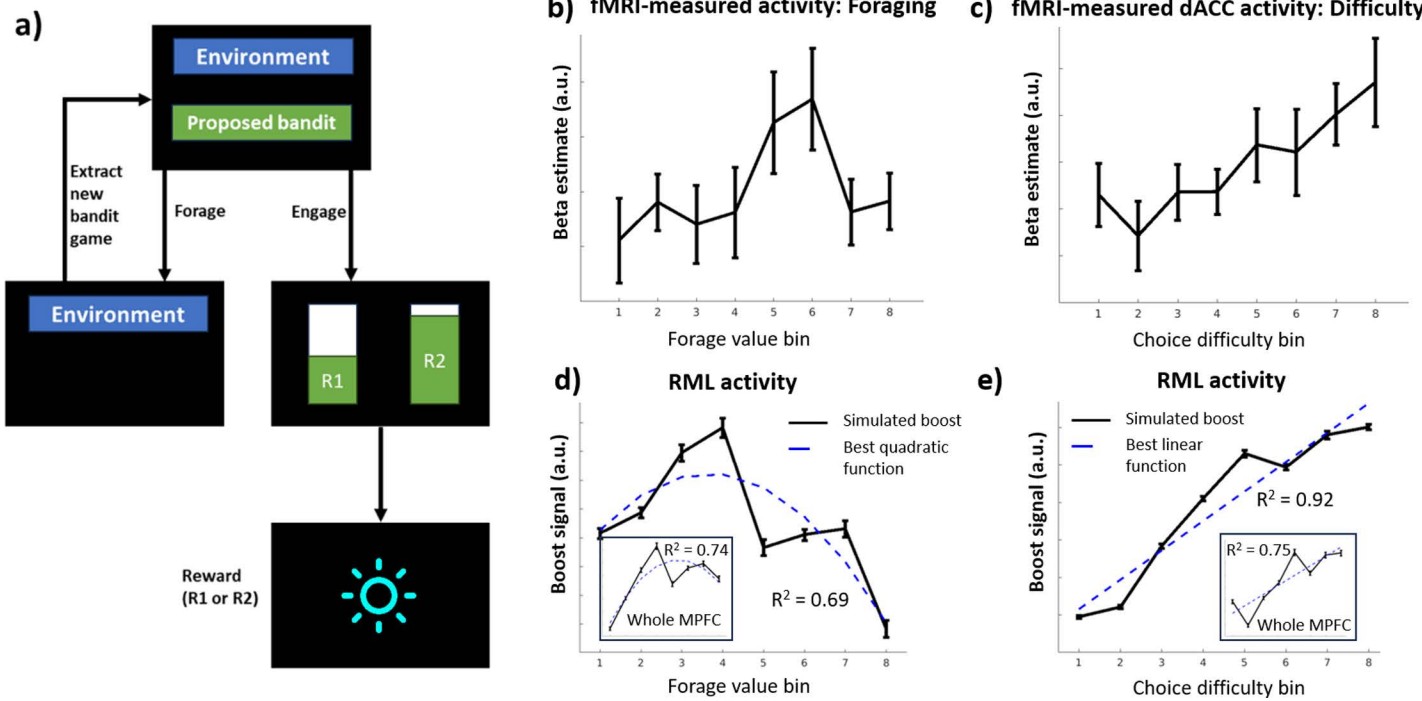

**Fig 5. Foraging task simulation. a.** *Single trial schema. Each trial began with a compound cue (e.g., green and blue) indicating the proposed bandit and its source environment. Choosing "engage" led to the next trial state where the indicated bandit was played. Choosing "forage" led to a waiting state (foraging cost), then back to the initial state with a new, randomly selected bandit from the same environment (green cue changes accordingly).* **b-c.** *fMRI results from* [17]*, showing respectively the dACC activity as a function of the value of the "forage" choice and as a function of the similarity between the "forage" and the "engage" options (choice difficulty).* **d.** *RML boost signal as a function of foraging value. The subplot shows the simulated activity of the whole MPFC sector of the RML, computed as the combination of boost and value signals.* **e.** *RML boost signal as a function of choice difficulty. As in d, the subplot shows the simulated activity of the whole MPFC, computed as the combination of boost and value signals.*

The expected net value component is dependent on the difference in cue value and follows a U-shaped function. When the values of both options are similar (around 0 on the x-axis in Fig 3D), the mean expected value of that trial type is minimal, while it is maximal for large value differences. Two mechanisms cause the shape of this function. First, similar option values lead to smaller drift rates in the DDM, and therefore to longer RTs (see also Fig D in S1 Text and Equations S10, S13). Since long RTs cause reward devaluation and a higher probability of exceeding the response deadline, the RML learns that the execution of this type of trial has a lower average value (minimum in the U-shaped pattern in Fig 3D). Conversely, when the difference between option values is large, RTs are shorter on average (higher drift rates), leading to a higher expected value (maxima in Fig 3D). The second mechanism depends on the intrinsic cost of boosting. The boost function mirrors the net value function (inverted U shape, see below), and the boost cost function – which is a linear function of boost - follows the same shape (S4b Equation in S1 Text). For this reason, the expected value is maximally discounted in the centre and minimally discounted in the tails, contributing to a U-shaped expected value. Fig 3E shows the contribution to the RML simulation of dACC activity given by the cognitive control signal (boost), which follows an inverted U-shaped function. As the boosting level maximizes reward while minimizing the cost intrinsic to boosting, the RML increases the boost only if the consequent reward gain is worth the boost cost. When the difference between the options is small, the DDM drift rate is small. In that case, a larger boost allows for a significant improvement in performance by shrinking the decision boundaries and therefore shortening the RTs (Fig 2A, see also S11 Equation in S1 Text). Conversely, when the task is easy, and the value difference is large, the DDM drift rate is high. In that case, RTs will be fast anyway, and increasing the boost will only marginally improve them. This leads to a reduced boost signal (cognitive control) when the value difference is large (minima in Fig 3E).

The RML predicts the W-shaped dACC activity equally well as the PRO model, which was successfully used in the original study [32]. Moreover, only the RML provides a further experimental prediction about another region of the MPFC: The vMPFC. Indeed, the value function in Fig 3D predicted the activity of this region (red cluster in Fig 3G), which is well known to be associated with value estimation [55,56]. Finally, the RML also predicts the dACC activity evoked by the reward-driven surprise computed as the absolute difference between the overall reward available in a trial and the long-term average reward (Fig 3F and green cluster in Fig 3G). This result is due to the surprise-coding mechanisms proper of both the RML and the PRO models.

**Working memory task**

In this simulation, we administered a verbal WM task -like the one used in Engström et al. [31] (Fig 4A)- to the RML. This task investigates the role of the dACC in cognitive effort in an experimental setting where surprise and prediction error are balanced across conditions so that they do not correlate with effort. The verbal WM task is an instantiation of the matching-to-sample paradigm, where a series of words are visually presented and, after a delay, the agent (either model or human) is asked to evaluate whether a target word belongs to the previously presented set. This experimental paradigm is well suited for investigating the optimization of the cognitive control signal (boost) in the RML, when task difficulty is not confounded with surprise.

In Fig 4B we show the dACC activity measured with fMRI during the WM task. As reported in Engstrom et al. [31], the dACC activity was significantly described by a linear function of WM load. The RML successfully predicted the dACC activity (Fig 4C) as boost intensity by the $MPFC_{Boost}$ module, showing –like in the fMRI results- a significant linear trend as a function of WM load (t-test on beta values: $t(14) = 17.58$, $p < 0.0001$). This result derives from the optimization of the boost signal as a function of task difficulty, to counteract –via LC output- the cRNN drop of performance due to increasing WM load.

The RML also showed that neither surprise (Fig 4D) nor value expectation (from both $MPFC_{Act}$ and $MPFC_{Boost}$, Fig 4E) can predict the dACC activity during the task (none resulted in showing a significant linear trend as a function of WM load). This was because accuracy remained constant across difficulty levels (mean accuracy 87%, main effect of WM load: $F(3,14) = 1.39$, $p = n.s.$) as reported in human participants [44].

**Foraging task**

In the last simulation, we tested the RML with a Foraging task (based on the tasks from Kolling et al. and Shenhav et al. [17,35], Fig 5A). Here the model was asked to perform a choice, in each trial, between engaging a two-armed bandit (exploit) or foraging for a new bandit game (explore), randomly extracted from the same context (patch). Both the context and the proposed bandit game were cued at the beginning of the trial. Before executing the task, the RML was trained to estimate both the average value of different contexts and of each bandit game.

Fig 5B and 5C shows the fMRI-related activity of the human dACC during the same task, as found by Shenhav et al. [17], revealing a non-monotonic response of the dACC as a function of the foraging value (Fig 5B) and a monotonically increasing dACC response as a function of choice difficulty (defined as the similarity in value between "engage" and "forage" options, Fig 5C). Shenhav et al. [17] reported these results as supporting the hypothesis that the human dACC activity codes for task difficulty rather than for foraging value, as instead proposed in Kolling et al. [35]. Indeed, in Shenhav et al. [17] it was shown that when the forage value is relatively low (corresponding to the range of values from Kolling et al. [35]), there is a linear increase of dACC activity as a function of the forage value. When the forage value becomes higher, however, the dACC activity decreases, suggesting that the forage-related activity attributed to the human dACC was due to the specific experimental settings.

The RML can simulate both the foraging value and the choice difficulty results from Shenhav et al. [17]. In Fig 5D, we show that the RML predicts a non-monotonic dACC activity as a function of the foraging value (Aikake weight for a quadratic vs linear function > 0.999). This activity pattern is driven by the boost optimization process and they hold both if

we consider the RML boost-related activity alone, and also if we combine all the signals from the MPFC modules of the RML (Fig 5D, and subplot). In this task, the boost signal is strongly influenced by the difference between the "engage" and "forage" values. When the value of the "forage" option is low, the relative difference in value between the two options is high and the choice in favor of the "engage" option is easy, as it is driven by the much higher expected value for engaging. When the foraging value increases, instead, the value difference between the two options decreases, making a quick and accurate selection of the optimal choice more difficult. This leads to an increase in the boost signal to optimize the decisions (activity peak in Fig 5D). However, increasing the foraging value even further makes the "forage" option more and more preferable with respect to the "engage" option, decreasing the choice difficulty, and thence the need for boosting (value bins > 4 in Fig 5D). To further investigate this effect, we analyzed also the RML activity as a function of choice difficulty (Fig 5E). When varying the choice difficulty, the RML finds a similar effect of choice difficulty on simulated dACC activity as Shenhav et al. [17]: we find that the boost activity linearly increases as a function of choice difficulty, and this activity pattern emerges also when considering the RML activity relative to the whole MPFC (Fig 5E subplot).

In summary, although the RML simulates the experimental results from both the Kolling et al. [35] and Shenhav et al [17] works, our model provides a novel computational explanation of these data, suggesting that the dACC activity in this type of tasks is neither driven by foraging value nor by choice difficulty per se, but rather by the optimization of cognitive control signals (boost) aimed at maximizing performance.

## Discussion

In this study, we proposed the meta-RL framework as a solution to the theoretical impasse generated by the fact that the two most prominent accounts of dACC in human cognition – namely, the performance monitoring framework and the EVC framework – focus on two distinct sets of studies, whose results are difficult to reconcile within a unified account. While the performance monitoring framework (instantiated with the PRO model) was effective in predicting the dACC activity in a cognitive control task involving unexpected events—the speeded decision-making task utilized in Vassena et al.[32]—it is ineffective in predicting the dACC activity in cognitive control tasks designed to exclude unexpected events (e.g., [3,24,29–31]). Differently, the EVC perspective could successfully frame the aspects of the dACC activity related to cognitive control optimization in the absence of surprise signals but does not account for the neural data when surprise and unexpected events are relevant for task execution [7]. At the same time, both theoretical frameworks are able to predict the activation of the dACC in a third type of experimental setting: The foraging task [35]. In this domain, the PRO and EVC frameworks both provided effective, but computationally different theories of the role played by the dACC in foraging tasks, one based on surprise coding (PRO, [36]), the other on choice difficulty coding [17], suggesting that the foraging task -at least as conceived by Kolling et al. and Shenhav et al. [17,35]- is inadequate to disentangle their predictions. All the analyzed fMRI studies led to overlapping clusters of activation centered on the dACC (Fig 6E), suggesting the central role of this areas in all the experimental paradigms, yet proposing different computational interpretations.

### Insights from RML simulations

Here we have shown that the meta-RL perspective (instantiated with the RML model [37,38]) can predict dACC activity in all these task types, as it grounds cognitive control optimization on Bayesian surprise tracking. Our two simulations showed that the meta-RL perspective can simultaneously account for findings that were previously explained separately by the EVC and PRO models. When applied to a speeded decision-making task that exemplifies the importance of surprise monitoring, the RML matched the PRO model in predicting the dACC activation (Fig 3C) and surprise-driven activity in the mid-cingulate cortex (Fig 3F), while also additionally predicting the vMPFC activity related to value estimation (Fig 3D), found by Vassena et al. [32] as a replication of experimental results from previous literature (e.g.,[55,56]). When applied to a working memory task that exemplifies the importance of cognitive effort, the RML successfully predicted dACC activity (Fig 4C) and the fact that neither surprise (Fig 4D) nor value expectation can predict the dACC activity

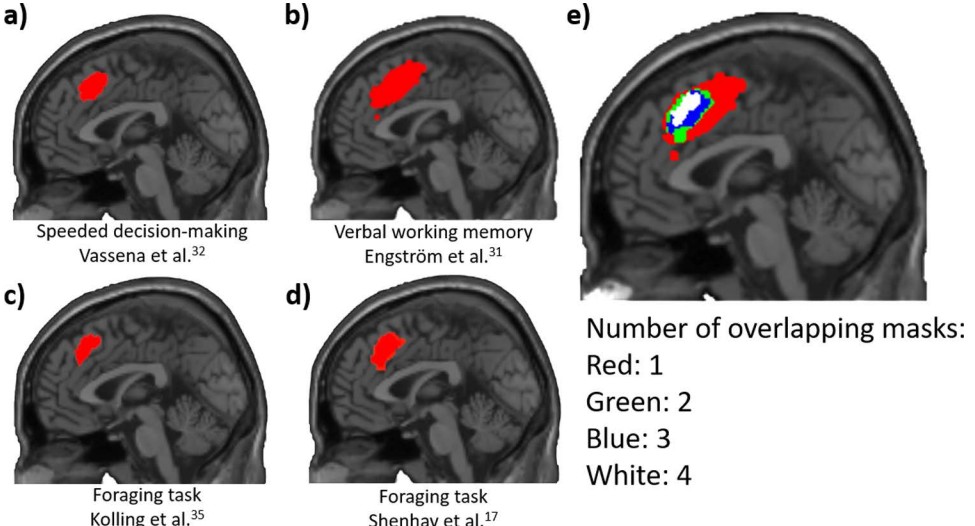

**Fig 6. Anatomical overlap of all the dACC fMRI clusters from the three experimental paradigms we simulated in this work, all plotted on a T1 image at X=0. a.** the cluster for the speeded decision-making task, based on data extracted from the figures from Vassena et al. [32]. **b.** the cluster for the verbal working memory task, based on data extracted from the figures from Engström et al. [31]. **c.** the cluster for the foraging task, based on data extracted from the figures from Kolling et al. [35]. **d.** the cluster for the foraging task, based on data extracted from the figures from Shenhav et al. [17]. **e.** the overlap between the four different clusters. The different colours indicate the number of overlapping clusters.

during the task (Fig 4E). Finally, when applied to a foraging task, the RML correctly predicts the foraging value (Fig 5D) and the choice difficulty (Fig 5E) coding in the human dACC.

It is important to note that while both the RML and PRO models generate comparable predictions regarding dACC activity in the speeded decision-making task [32], they provide distinct underlying computational explanations. The PRO model generates the W-shaped dACC activity pattern as the combination of two different types of surprise signals. One surprise signal is triggered by the cue onset, and it depends on the uncertainty relative to the cue type. The other is triggered by the response onset and depends on the uncertainty related to the probability of selecting one of the two responses over the other. The PRO model therefore suggests that the dACC plays no role in the optimization of cognitive control per se but it is exclusively involved in surprise tracking. As argued above, this view is challenged by empirical evidence showing dACC activation corresponding to cognitive control intensity in tasks deliberately designed to exclude surprise [3,24,29–31]. In contrast, the RML explains the dACC activity as the combination of two different functions: The expected value of the selected action (U-shaped function in Fig 3D) and the cognitive control intensity (inverted U-shaped function in Fig 3E), suggesting a more complex function of the dACC, where surprise-based monitoring is functional to cognitive control optimization. These results are supported also by cognitive neuroscience works (e.g., [57,58]), which suggest a modular organization of the MPFC, with the ventromedial part coding mainly for value and cost, while the dorsal part coding mainly for effort mobilization (see activation clusters in Fig 3G), and also by neurophysiological works, reporting that in the dACC regions, there are neural populations coding for both monitoring and effort-related information [59,60].

This is the reason why the RML, differently from the PRO model, is able to predict the dACC activity also in tasks where surprise is not involved, such as the working memory task studied here (see also [37,38,41]). These findings solidify the RML as a computational account unifying the monitoring and control perspective during decision-making, thereby reconciling surprise, value, and cognitive control under the framework of meta-learning processes [39,40].

The RML effectively predicted also the dACC activity during the foraging task, and also in this case it provided a different computational account of the function of this area, compared to both the EVC and the PRO models. Previous works

have shown that while the PRO model predicted the dACC activity during the foraging task as mostly linked to surprise [36], the EVC framework pointed out that the activation pattern of the dACC in the same task could be explained as due to the difficulty of the choice between the "forage" and "engage" options. Differently, the results from RML simulations propose that the role of the dACC during foraging is about the optimization of resources that drives the arbitration between the decision to forage or to exploit the resources available in the current patch.

Another important innovation of the RML is its bio-inspired system-level perspective, where MPFC function is studied in conjunction with brainstem and midbrain catecholaminergic nuclei. This allows for modeling of the cortico-subcortical reciprocal influence and how the MPFC can orchestrate cortical and striatal functions by controlling the release of neuromodulators, broadening the scope of application of the RML framework. For example, in previous work, the RML was shown to be capable of providing a computational account of how the MPFC modulates parietal neurons - via LC activation - during a visual attention task [38]. The same mechanism can also explain the activity in dorsolateral regions of the frontal and parietal lobes that Vassena et al. described in their study [32]. Indeed, these areas are involved in the execution of the speeded decision-making task (spatial and motor control components), and the RML suggests that the MPFC optimizes their performance by neuronal gain modulation via neuromodulatory input. Moreover, this system-level perspective allows us to formulate predictions about different regions within the MPFC, like the activity pattern of the vMPFC (Fig 3D) or of the mid-cingulate cortex (Fig 3F).

A further relevant element introduced by the RML is that most of the behavioural and neural predictions made by the RML emerge from the interaction of the model with the environment (*situatedness* as defined by Wilson [61] and Nolfi [62]). Furthermore, the goal of the optimization process of the RML is to find the set of actions (policy) that maximizes the net value, regardless of whether these actions are directed toward the external environment (motor) or the internal environment (cognitive control). Learning optimal actions (motor or cognitive) requires a loop between the RML and the environment, where the RML plays an *active* role in exploring the world. Recently, an improvement to the RML was proposed, the RML-C [38], where the situated and proactive aspects of RML are more pronounced, as the model's goal is not limited to net value maximization, but includes also the maximization of information relative to the environment. To this aim, the RML-C actively explores the environment to improve its predictions, investing motor and cognitive effort for gathering information (*intrinsic motivation* [63,64]), beyond the utilitarian perspective of net value maximization. The situated and proactive perspective that the RML proposes about the dACC role – and more in general of the MPFC – seems to be soundly grounded also on the neurophysiology of this area, which evidences large neural populations coding for both motor and visceromotor functions (for a review: [26]). Differently from the RML, the PRO model embraces a pure monitoring perspective, where the dACC plays a passive role of RL "critic" [33,65], dedicated to the monitoring of action-outcome contingencies. Similarly, the implementation of the EVC model proposed by Vassena et al. [32] selects the control intensity as a function of the option values and the control cost, working as a decision-making algorithm where information about the environment and the action-outcome contingencies is fully known. A more recent implementation of the EVC, the Learned Value of Control (LVOC) model [28], has been designed for learning the optimal policies of control through model-environment interactions in stationary conditions. However, contrary to the RML and PRO models, the LVOC does not include explicit monitoring of surprise, hence it remains to be tested whether it could account for the dACC data relative to surprise coding.

### Limitations and future directions

We identify four main limitations in our approach, which we discuss here together with the description of future works aimed at overcoming them. The first concerns the need for a replication of the experimental findings relative to the speeded decision-making task by Vassena et al. [32]. This was the first study using a task designed ad hoc to differentiate dACC activation predictions from performance monitoring and cognitive control frameworks, and led to a debate over the actual involvement of cognitive control processes in the speeded decision making task [66,67]. We believe that further evidence confirming the "W-shaped" dACC activation pattern in this task would certainly strengthen Vassena et al.'s [32]

results. However, this limitation has little impact on our work, as the theoretical impasse generated from the comparison of the performance monitoring and the cognitive control frameworks emerges independently of the results from the speeded decision-making task. This problem arises also from the analysis of previous studies (some cited in the Introduction) where different signals related to surprise, cognitive control, choice difficulty, etc., were independently detected in the dACC. The second limitation is that we did not examine the role of intrinsic motivation and information gathering in our model's utility function. The simulations presented here concern the RML model, which optimizes effort based on extrinsic expected value (reward). Yet, recent studies show that intrinsic motivations aimed at minimizing uncertainty also influence MPFC function [38]. As noted above, our latest model, RML-C, incorporates this aspect. In this work, we chose RML over RML-C because intrinsic motivation was not a key factor in the experimental settings considered, making RML sufficient to support our theory. To verify this, we additionally tested the RML-C (results in S1 Text), which made a similar prediction to the predicted dACC activity by the RML in the speeded decision-making task. Future studies should further explore the advantages of RML-C in predicting neural and behavioral data in more ecological settings, where intrinsic and extrinsic motivation interact (e.g., Silvetti et al. [38]). The third limitation deals with the exact role played by the brainstem neuromodulatory nuclei in MPFC-driven effort and motivation control. Although we based the design of the RML on neuroscientific literature, e.g., about the MPFC-brainstem connections and the role of LC in effort (see Silvetti et al., 2018, 2023 for a literature survey and the limitations of our modeling of catecholamines [37,38]), there are no studies yet that provide direct evidence for the roles of LC and VTA as hypothesized by the RML. For this reason, a direct test of the assumptions of the roles of LC and VTA is still pending (e.g., through systematic pharmacological challenge studies), representing an important step in future research. Finally, the fourth limitation comes from discrete Markov decision process (MDP) approach we adopted to implement the RML (see also Supplementary Methods in S1 Text). Although this choice effectively reduced model complexity (in Silvetti et al., 2018 we also provided a continuous MDP version of the model with great cost in complexity [37]) it also reduced the time resolution of our predictions, making relatively difficult to generate testable predictions about the exact timing of MPFC activity in different tasks. Nonetheless, the problems we faced in this work indicate that there are still crucial questions to be answered and that they can be still effectively addressed by the simplified discrete MDP approach.

In summary, in this work, we have shown that the meta-learning perspective may represent a general solution for the understanding of the MPFC function and that it is capable of predicting empirical data from a larger domain set if compared with the PRO and EVC models alone. Future research in this domain should investigate also the comparison of the above frameworks with other, alternative perspectives on the computational basis of cognitive control, based on active inference [68,69], such as the proposal that cognitive control results from the monitoring of the deviation from prior beliefs about cognitive actions so that cognitive effort is exerted to override habits [70].

## Supporting information

**S1 Text. Supplementary methods and results.**
(DOCX)

## Acknowledgments

TV is a PhD student enrolled in the National PhD in Artificial Intelligence, XXXVII cycle, course on Health and life sciences, hosted by Università Campus Bio-Medico di Roma.

## Author contributions

**Conceptualization:** Tim Vriens, Massimo Silvetti.

**Data curation:** Tim Vriens, Massimo Silvetti.

**Formal analysis:** Tim Vriens, Massimo Silvetti.

**Funding acquisition:** Massimo Silvetti.

**Investigation:** Tim Vriens, Massimo Silvetti.

**Methodology:** Tim Vriens, Eliana Vassena, Giovanni Pezzulo, Gianluca Baldassarre, Massimo Silvetti.

**Project administration:** Massimo Silvetti.

**Resources:** Massimo Silvetti.

**Software:** Tim Vriens, Massimo Silvetti.

**Supervision:** Massimo Silvetti.

**Visualization:** Tim Vriens, Massimo Silvetti.

**Writing – original draft:** Tim Vriens, Eliana Vassena, Giovanni Pezzulo, Gianluca Baldassarre, Massimo Silvetti.

**Writing – review & editing:** Tim Vriens, Eliana Vassena, Giovanni Pezzulo, Gianluca Baldassarre, Massimo Silvetti.

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
