## [Decision Letter · Decision Letter 0]

29 Oct 2024

PCOMPBIOL-D-24-01531Meta-Reinforcement Learning reconciles surprise, value, and control in the anterior cingulate cortex.PLOS Computational Biology Dear Dr. Silvetti, Thank you for submitting your manuscript to PLOS Computational Biology. After careful consideration, we feel that it has merit but does not fully meet PLOS Computational Biology's publication criteria as it currently stands. Therefore, we invite you to submit a revised version of the manuscript that addresses the points raised during the review process. As you can see from the below reviews, all reviewers liked the manuscript and highlighted its merit. They also made a number of suggestions on how to make the paper more accessible and bring in conflicting views in a balanced way. We ask you to implement these suggestions in a major revision, Please submit your revised manuscript within 60 days Dec 29 2024 11:59PM. If you will need more time than this to complete your revisions, please reply to this message or contact the journal office at ploscompbiol@plos.org.  Please include the following items when submitting your revised manuscript: * A rebuttal letter that responds to each point raised by the editor and reviewer(s). You should upload this letter as a separate file labeled 'Response to Reviewers'. This file does not need to include responses to formatting updates and technical items listed in the 'Journal Requirements' section below.* A marked-up copy of your manuscript that highlights changes made to the original version. You should upload this as a separate file labeled 'Revised Manuscript with Track Changes'.* An unmarked version of your revised paper without tracked changes. You should upload this as a separate file labeled 'Manuscript'. If you would like to make changes to your financial disclosure, competing interests statement, or data availability statement, please make these updates within the submission form at the time of resubmission. Guidelines for resubmitting your figure files are available below the reviewer comments at the end of this letter. We look forward to receiving your revised manuscript. Kind regards, Tobias U Hauser, PhDAcademic EditorPLOS Computational Biology Hugues BerrySection EditorPLOS Computational Biology Feilim Mac GabhannEditor-in-ChiefPLOS Computational Biology Jason PapinEditor-in-ChiefPLOS Computational Biology  **Journal Requirements:**  **Additional Editor Comments (if provided):****Reviewers' comments:**  Reviewer's Responses to Questions

**Comments to the Authors:**

Reviewer #1: In their study titled “Meta-Reinforcement Learning reconciles surprise, value, and control in the anterior cingulate cortex.”, Vriens et al. compare the predictions of meta-reinforcement learning (MRL) model simulations to actual signals recorded in the dorsal anterior cingulate cortex (dACC). The topic is timely, as several incompatible theories exist, and a consensus is still lacking (see Clairis & Lopez-Persem, 2023 for a full explanation of these theoretical conflicts). Therefore, studies like this one are necessary in the field of cognitive neuroscience.

In their article, the authors found that predictions from their model, which had previously been introduced in several publications, matched the dACC signals recorded during two tasks: a speeded binary choice task and a verbal working memory task. While the methods are robust and the findings are interesting, I believe that more in-depth analysis is required to fully support their main claim—that they resolve the theoretical impasse surrounding the function of the dACC (this claim should be alleviated). Some relevant literature, particularly studies that conflict with their framework, is not covered sufficiently. Despite these weaknesses, I believe the manuscript can be significantly improved with major revisions.

Theoretical aspects of the manuscript (introduction and discussion):

1. In the abstract and introduction, the authors state that two competing frameworks, the EVC and the performance monitoring framework, attempt to find a unified theory of dACC function. However, other studies suggest that the dACC function is also related to foraging value (see Clairis and Lopez-Persem 2023 for an overview of these theories). To be more inclusive, I recommend that the authors briefly integrate this literature (Kolling and colleagues work) into both the introduction and discussion.

2. In the introduction, the authors state: “Furthermore, a recent fMRI study compared the predictions of the two frameworks during a speeded decision-making task, which involved both cognitive control and performance monitoring32.” Indeed, but this was challenged by Shenhav A, Musslick S, Botvinick MM, Cohen JD. Misdirected vigor: Differentiating the control of value from the value of control. PsyArXiv. [Preprint]. 2020. doi:10.31234/osf.io/5bhwe. Could the authors address the criticisms raised in this commentary and explain how they dissociate control from vigor in their manuscript?

3. The article from Silvetti et al, 2018 (ref 34 in the manuscript) already attempts to demonstrate that the RML is efficient in explaining the dACC function. Can the authors clarify the novelty of their findings in comparison to previous work?

4. From the discussion, I understand that RML is not the most recent model, so why not use RML-C?

On the choice of experimental designs:

5. The article would be much more convincing if simulations were conducted on additional experimental designs that are known to trigger dACC activity. For instance:

• Why simulate a simplified version of the choice task from Vassena et al.? What are the results of the actual task from Vassena et al.?

• How does the model perform on tasks designed by Kolling et al., which show that the dACC encodes foraging value?

• What are the model's predictions for tasks involving effort (such as those in Clairis and Pessiglione 2022, 2024)?

Results:

6. One of the most consistent findings in binary choice tasks is that the dACC encodes the value of the unchosen option positively and the value of the chosen option negatively. Consequently, the decision value (Chosen-Unchosen) is coded negatively. In this paper, the authors show results from a previous fMRI experiment and simulations depicting a value difference in the side frame (i.e., right vs. left option values). It seems that what they call a quartic relationship might actually correspond to a quadratic relationship in terms of decision value (DV, choice frame) with the dACC signal. I have several questions and suggestions in this regard:

o Can the authors justify their choice of the side frame?

o To my knowledge, only one article demonstrates this quartic relationship. Many datasets with binary choice tasks are available on open platforms. It would be reassuring if the authors could demonstrate that this quartic relationship holds in other datasets.

o How sensitive is this quartic relationship to time pressure? What are the model's predictions without time pressure?

o What are the model’s predictions regarding the encoding of chosen and unchosen option values?

o A low DV can correspond to high-high or low-low options. Does the model make different predictions for these two cases? One might expect that high-high options require less "effort," as the consequences of both actions are positive.

o What are the model’s predictions regarding response times?

7. Regarding the anatomical definition of the dACC: the brain regions involved in the speeded decision-making task and the verbal WM tasks are not the same. One difficulty in establishing a consensus on dACC function is to clearly delineate which brain region is being studied. Is the signal in the dACC, as defined by the ROI in the WM task, similar to the one identified in the speeded decision-making task? Is the reverse also true?

Minors:

Terminology: Rather than stating that the MRL "can explain" the dACC function, it would be more accurate to say that it "can mimic" or "resembles" the dACC function, as the results are based on simulations.

Reviewer #2: This manuscript presents the results of two simulations using a reinforcement meta-learner (RML) model to model the activity of the dorsal anterior cingulate cortex and compare it with published fMRI results. The aim is to test the hypothesis that this brain region integrates outcome/performance/surpise monitoring and cognitive control. While this is not a novel hypothesis, the authors are correct in stating that previously proposed mathematically formalized models are usually capable to explain only one of these two aspects. The RML model is applied to a speeded decision making task and a working memory task and captures the neuroimaging results by predicting both cognitive control and monitoring functions.

This is an interesting and well-written manuscript. The methods are sound and well-explained in the supplementary materials. I have only very few and minor comments that may help optimizing the contribution of this manuscript to the literature.

1. The RML model is stated to be bioinspired and individual modules are linked to certain brain regions. For the dACC (MPFC) this is obvious and needed to create the link to the fMRI results. Also the involvement of the LC and VTA appear reasonable. However, it might be necessary to state that a direct test of the assumptions of the roles of LC and VTA are still pending (for example, through systematic pharmacological challenge studies). Also, as a minor remark, I was surprised that only LC has an inflience of the decision boundary in the DDM, given that dopamine has a clear effect on response vigor.

2. I thin that some further elaboration on the putative common representation of cognitive control and monitoring variables in dACC would be helpful. Are these two sets of variables and computations assumed to be carried out by the same neuronal ensembles? It might be worth discussing this in the light of findings that representations in the dACC and adjacent regions are multiplexed (e.g., Fu et al., Science, 2022; Kennerley et al., 2011, Nat Neurosci). Also, it might be important to point out the timing of the two functions. While in fMRI studies the different activity increases related to monitoring and control, respectively, seem to overlap in a common activity pattern, EEG, MEG, and intracranial recordings can reveal a temporal dissociation. Can the RML make clear and testable predictions for time-resolved studies?

3. very minor: Figure caption in Fig 3 refers to panel g which does not exist.

Reviewer #3: In this manuscript, Vriens et al. compare simulations from a Meta-Reinforcement Learning model to fMRI data from two tasks that involve cognitive control. These tasks are selected because results could not be explained by a single existing model. In contrast, the Reinforcement Meta-Learner (RML) model successfully captures neuroimaging results for both tasks. The authors focus on activity in dorsal anterior cingulate cortex (dACC), as the role of this region has been particularly debated in previous work, and activity patterns are well explained by the RML model.

The work addresses important questions about the neural and computational basis of cognitive control and the role of dACC specifically. Providing evidence for a model that unifies previously conflicting findings is an important advance in the field. The modelling seems robust, with additional control or replication analyses provided. I have some suggestions that I hope are helpful to further improve the manuscript.

1. The authors present convincing visual evidence that the RML simulations closely capture the dACC activity patterns from the original fMRI results in both studies. They additionally provide statistical evidence that the same pattern (quartic, linear) best fits both the fMRI measured and RML simulated activity. I wondered whether there are more direct ways to show that the RML simulations are the best fitting model of the fMRI activity? It would also be helpful if possible to include fit statistics to show the badness or lack of fit in the relevant places, for example surprise and expected value for the verbal VM task, potentially using Bayesian statistics to show a lack of association.

2. While the findings shown in the figures are convincing, the figures and legends could be improved:

a. The text on Figure 3a, 3b etc is very small

b. It might be helpful to combine Figure 3a with 3b and combine Figure 5a with 5b, even if separate y axes are required, to show how well the simulations capture activity patterns more directly. This point seems more relevant to the main argument of the paper than the fit of the quartic functions for Figure 3. If the authors decide to keep the quartic models in Figure 3, the linear trends could be added to Figure 5 for consistency.

c. It is unclear what the meaning or interpretation of Figure 3e is.

d. The figure legends would benefit from additional detail throughout. Figure 3 and 5 specifically would also be easier to interpret with links to the main findings for example that they support the RML model, surprise is not sufficient to capture pattern etc.

3. I think changes to the structure of sections in the manuscript could help highlight the key contributions of the work. The results section contains a lot of methodological information and the methods section at the end refers to the supplementary information. It seems it may be possible instead to have a methods section between the introduction and results with the sections describing the model and tasks. Then the results section can get into the key findings much sooner.

4. In addition to the structure, the writing could also better highlight the key findings in a way that is easier for readers less familiar with the exact predictions of different models. For example, starting results sections with the specific aim of that analysis, relative to the overall aim of the paper, and using ‘signposting’ phrases such as “As expected if the RML model captures… we found...”.

5. The number of participants simulated for the speeded task or whether results are averaging across participants is not clear.

6. There seems to be inconsistencies between the main text and supplement on whether there are 3 components in 2 modules or 3 modules. If some quantifications are specific to MPFC activity or another subdivision this could be clearer.

7. The authors clearly acknowledge that equating the boost module in the model with cognitive control is a simplification, could the discussion include a short section on the implications of this?

8. The discussion highlights the RML-C as an improvement to the RML. Could this section also state whether the differences between these versions of the model would impact the findings of the current paper? This could include suggesting how future work might examine the association between intrinsic motivation and cognitive control, based on the importance of task design and selection as highlighted in this work.

Minor:

1. The following sentence is very long, splitting it would help clarity: “The dorsal anterior cingulate cortex (dACC) and the surrounding cortical areas within the medial prefrontal cortex (MPFC) are known to be involved in cognitive control processes (e.g. 6), as well as in Reinforcement Learning (RL) and decision-making (e.g. 7), and the underlying computational mechanisms are still unclear, above all considering that their activation has been routinely found in many different experimental paradigms investigating several aspects of higher-order cognition8–19”

2. As this section is comparing the two existing frameworks, the “however” in the middle is a bit confusing as it comes before further evidence supporting EVC. “A growing body of literature reports activity in the dACC linked to cognitive control (or more generally during anticipation of cognitive effort), in keeping with the perspective of the EVC framework21 (see also 27,28). However, given that in some of these cognitive control studies dACC activity is observed in the absence of surprise or prediction errors3,24,29 or while controlling for all possible sources of surprise30,31, these findings are not easily accommodated by the performance monitoring framework.”

**Have the authors made all data and (if applicable) computational code underlying the findings in their manuscript fully available?**

Reviewer #1: Yes

Reviewer #2: Yes

Reviewer #3: Yes

PLOS authors have the option to publish the peer review history of their article (what does this mean? ). If published, this will include your full peer review and any attached files.

**Do you want your identity to be public for this peer review?** For information about this choice, including consent withdrawal, please see our Privacy Policy .

Reviewer #1: No

Reviewer #2: No

Reviewer #3: No

  **Figure resubmission:** While revising your submission, please upload your figure files to the Preflight Analysis and Conversion Engine (PACE) digital diagnostic tool, https://pacev2.apexcovantage.com/. PACE helps ensure that figures meet PLOS requirements. To use PACE, you must first register as a user. Registration is free. Then, login and navigate to the UPLOAD tab, where you will find detailed instructions on how to use the tool. If you encounter any issues or have any questions when using PACE, please email PLOS at figures@plos.org. Please note that Supporting Information files do not need this step. If there are other versions of figure files still present in your submission file inventory at resubmission, please replace them with the PACE-processed versions. 
---

## [Decision Letter · Decision Letter 1]

4 Apr 2025

Dear Dr. Silvetti,

We are pleased to inform you that your manuscript 'Meta-Reinforcement Learning reconciles surprise, value, and control in the anterior cingulate cortex.' has been provisionally accepted for publication in PLOS Computational Biology. As you can see from the reviewers' final comments, all of them acknowledged your thorough revisions and are happy with the paper as it stands.

Best regards,

Tobias U Hauser, PhD

Academic Editor

PLOS Computational Biology

Hugues Berry

Section Editor

PLOS Computational Biology

Reviewer's Responses to Questions

**Comments to the Authors:**

Reviewer #1: The authors have carefully addressed all my comments and substantially revised their manuscripts. I have no further comments and I am looking forward to seeing the manuscript published.

Reviewer #2: By an extensive revision of the article and the inclusion of a foraging task the authors substantially improved the clarity and impact of the manuscript. In my view, all reviewers' comments were addressed in the revision. I have no further suggestions or comments.

Reviewer #3: Thank you to the authors for their thorough response to my comments and other revisions to the manuscript - replicating the findings on a third task is particularly impressive. I think this work makes an important contribution to the field.

**Have the authors made all data and (if applicable) computational code underlying the findings in their manuscript fully available?**

Reviewer #1: None

Reviewer #2: Yes

Reviewer #3: Yes

PLOS authors have the option to publish the peer review history of their article (what does this mean? ). If published, this will include your full peer review and any attached files.

**Do you want your identity to be public for this peer review?** For information about this choice, including consent withdrawal, please see our Privacy Policy .

Reviewer #1: No

Reviewer #2: No

Reviewer #3: No

---

## [Editor Report · Acceptance letter]

PCOMPBIOL-D-24-01531R1

Meta-Reinforcement Learning reconciles surprise, value, and control in the anterior cingulate cortex.

Dear Dr Silvetti,

I am pleased to inform you that your manuscript has been formally accepted for publication in PLOS Computational Biology. Your manuscript is now with our production department and you will be notified of the publication date in due course.

With kind regards,

Lilla Horvath

PLOS Computational Biology | Carlyle House, Carlyle Road, Cambridge CB4 3DN | United Kingdom ploscompbiol@plos.org | Phone +44 (0) 1223-442824 | ploscompbiol.org | @PLOSCompBio